# Hypothyroidism-Induced Nonalcoholic Fatty Liver Disease (HIN): Mechanisms and Emerging Therapeutic Options

**DOI:** 10.3390/ijms21165927

**Published:** 2020-08-18

**Authors:** Daniela Maria Tanase, Evelina Maria Gosav, Ecaterina Neculae, Claudia Florida Costea, Manuela Ciocoiu, Loredana Liliana Hurjui, Claudia Cristina Tarniceriu, Mariana Floria

**Affiliations:** 1Department of Internal Medicine, “Grigore T. Popa” University of Medicine and Pharmacy, 700111 Iasi, Romania; tanasedm@gmail.com (D.M.T.); floria.mariana@umfiasi.ro (M.F.); 2Internal Medicine Clinic, “Sf. Spiridon” County Clinical Emergency Hospital Iasi, 700115 Iasi, Romania; 3Department of Gastroenterology, “Grigore T. Popa” University of Medicine and Pharmacy, 700115 Iasi, Romania; ecaterina.neculae91@gmail.com; 4Institute of Gastroenterology and Hepatology, “Sf. Spiridon” County Clinical Emergency Hospital Iasi, 700111 Iasi, Romania; 5Department of Ophthalmology, “Grigore T. Popa” University of Medicine and Pharmacy, 700115 Iasi, Romania; claudia.costea@umfiasi.ro; 62nd Ophthalmology Clinic, “Prof. Dr. Nicolae Oblu” Emergency Clinical Hospital, 700115 Iași, Romania; 7Department of Pathophysiology, Faculty of Medicine, “Grigore T. Popa” University of Medicine and Pharmacy, 700115 Iasi, Romania; manuela.ciocoiu@umfiasi.ro; 8Department of Morpho-Functional Sciences II, Physiology Discipline, “Grigore T. Popa” University of Medicine and Pharmacy, 700115 Iasi, Romania; loredana.hurjui@umfiasi.ro; 9Hematology Laboratory, “Sf. Spiridon” County Clinical Emergency Hospital, 700111 Iasi, Romania; 10Department of Morpho-Functional Sciences I, Discipline of Anatomy, “Grigore T. Popa” University of Medicine and Pharmacy, 700115 Iasi, Romania; claudia.tarniceriu@umfiasi.ro; 11Hematology Clinic, “Sf. Spiridon” County Clinical Emergency Hospital, 700111 Iasi, Romania; 12Internal Medicine Clinic, Emergency Military Clinical Hospital, 700483 Iasi, Romania

**Keywords:** nonalcoholic fatty liver disease (NAFLD), hypothyroidism, thyromimetics

## Abstract

Nonalcoholic fatty liver disease (NAFLD) is an emerging worldwide problem and its association with other metabolic pathologies has been one of the main research topics in the last decade. The aim of this review article is to provide an up-to-date correlation between hypothyroidism and NAFLD. We followed evidence regarding epidemiological impact, immunopathogenesis, thyroid hormone-liver axis, lipid and cholesterol metabolism, insulin resistance, oxidative stress, and inflammation. After evaluating the influence of thyroid hormone imbalance on liver structure and function, the latest studies have focused on developing new therapeutic strategies. Thyroid hormones (THs) along with their metabolites and thyroid hormone receptor β (THR-β) agonist are the main therapeutic targets. Other liver specific analogs and alternative treatments have been tested in the last few years as potential NAFLD therapy. Finally, we concluded that further research is necessary as well as the need for an extensive evaluation of thyroid function in NAFLD/NASH patients, aiming for better management and outcome.

## 1. Introduction

Nonalcoholic fatty liver disease is the most frequent cause of chronic liver disease that affects millions of people worldwide [1,2]. Genetics, obesity, unhealthy lifestyle, and other metabolic risk factors could be responsible for the burgeoning evolution, increased prevalence, and incidence of NAFLD. This multifactorial and dynamic entity is defined by hepatic fat accumulation in the absence of hereditary and autoimmune conditions, drug-induced liver injury, alcohol consumption, or viral etiology [3,4]. NAFLD encompasses a wide range of liver conditions from simple steatosis/nonalcoholic fatty liver (NAFL) to nonalcoholic steatohepatitis (NASH), cirrhosis, and finally, hepatocellular carcinoma (HCC) [5]. Considering the common and possible evolution to NASH and cirrhosis, NAFLD has become one of the main causes of liver transplantation [6]. It is characterized by comorbid manifestations that extend beyond the liver and in an all-cause mortality/morbidity relationship, plenty of evidence highlights its association with metabolic disorders, type 2 diabetes mellitus (T2DM), chronic kidney disease (CKD), gallstone disease, cardiovascular, and endocrine illnesses (hypothyroidism, polycystic ovarian syndrome, or hypogonadism) [7,8,9,10,11,12]. As NAFLD is considered the cause or the consequence of metabolic syndrome, which is reflected primarily by raised visceral adiposity that promotes insulin resistance (IR), glucose intolerance, and overall a lipotoxic status, it is possible that its metabolic substrates may share pathogenic factors with hypothyroidism [13,14,15,16]. More than that, below range serum concentration of thyroid hormones (THs) has been linked to the development of metabolic syndrome, obesity, atrial fibrillation, and cancer onsets [17,18,19,20]. 

Hypothyroidism continues to remain a global problem with increased incidence among the adult and newborn population, and illustrates decreased metabolic rate determined by hyposecretion of THs from the thyroid gland [21]. Studies have shown that patients with over ten years of thyroid dysfunction have significantly higher chances of developing HCC [22], and that in subjects with NASH and chronic hepatitis B infection, a higher thyroid dysfunction was found compared to that of the control group [23]. Hence, hypothyroidism may probably be the best example of NAFLD secondary to an endocrine disorder. Notably, hypothyroidism-induced NAFLD (HIN) was acknowledged recently as a distinct disease entity [24]. More than that, treatment of hormonal deficiency has been shown to improve liver condition [25] and that hyperthyroidism may improve NASH [26], which raises the interesting question of whether hormone replacing therapy could be beneficial in NAFLD.

In this review, we discuss the growing epidemiological evidence associating these two diseases, the key factors behind immunopathogenesis, how hormone replacing therapy impacts liver function, which of the newest therapeutic targets are being researched, and mostly, we hope to raise awareness among clinicians about the possible liver impairment in endocrine pathologies as well as the need for the evaluation of thyroid function in NAFLD/NASH patients for better management and outcome [24].

### Definitions and Epidemiological Discoveries

For many years, scientific research has been conducted in order to demonstrate an association and/or a causal relationship between hypothyroidism and NAFLD. Before we briefly discuss the epidemiological grounds of HIN, we must mention the clinical and biochemical definition of hypothyroidism and NAFLD.

Hypothyroidism is characterized by high plasma thyroid-stimulating hormone (TSH) in concurrence with either low plasma TH levels and a free thyroxine (fT4) that is within the reference range without obvious clinical symptoms (subclinical hypothyroidism) or with low fT4 accompanied by obvious clinical symptoms (overt hypothyroidism). Related to the cause, hypothyroidism is primarily due to thyroid dysfunction and secondary to the flawed hypothalamic–pituitary axis or to different disorders [27,28]. Current guidelines recommend that for subclinical hypothyroidism diagnosis, the cut-off level of TSH used should be between 4.0 and 4.5 mIU/l. Some evidence has shown that the upper normal limit of TSH interpreted by laboratory tests depends on many variables and conditions [29]. On the other hand, hepatic steatosis is defined by liver fat accumulation over 5% of hepatocytes with the lack of other secondary causes or by over 5.6% of proton density fat detected by magnetic resonance imaging (MRI)/spectroscopy [4,5]. Nonetheless, the golden standard for diagnosis remains the invasive liver biopsy, despite its potential severe side effects [4,30]. These definitions are very relevant to our later debate and might explain the confuting results found in studies that have investigated the entangled associations between hypothyroidism and NAFLD.

In the last years, substantial growth of inflammatory and endocrinological diseases has also been observed in a younger population. Hyperthyrotropinemia is defined as normal T4 levels accompanied by abnormal transient rise of TSH levels after the first two days of life, which goes back to normal after two or three weeks [31]. The intriguing transient hyperthyrotropinemia found in many neonatal children could be a consequence, or perhaps could play an independent part in the later adult development of obesity, IR, and dyslipidemia [32]. The congenital hypothyroid pediatric population displayed a positive relationship between TSH and lipid values and a higher prevalence of NAFLD [33]. Data showed that children with obesity and dysregulation of TH levels have a higher risk of developing hepatic steatosis [34]. In euthyroid overweight and obese children, researchers found a prevalence of NAFLD of almost 30% and that NAFLD individuals had significantly higher TSH levels than non-steatosis subjects, while T3 and T4 levels had similar values in both groups [35]. Moderate thyroid dysfunction could contribute to an unfavorable metabolic status in an overweight and obese younger population, which underlines the possible role of subclinical hypothyroidism as a predictive marker of metabolic dysfunction and open new views about possible metabolic therapeutic targets that could benefit both disorders [36].

Concerning the evaluation of HIN among the adult population, substantial evidence has emerged over the last two decades and includes cross-sectional [37,38,39,40,41,42,43,44,45,46,47,48,49], case-control [37,43,50,51,52,53,54], cohort [55,56], retrospective analysis [57], and meta-analyses studies [57,58,59,60]. Among these, some found a higher prevalence of NALFD among hypothyroid patients (subclinical and overt hypothyroidism) in a dose-dependent manner [37,39,42,43,45,50,52,53,59,60,61], a linear association between TSH levels and increased risk of NALD/NASH [35,40,44,45,61], and a correlation between higher fT3 [46,56,59] or low fT4 [41,51,56,59,62] plasma levels with raised risk of liver steatosis. Additionally, fT3/fT4 ratio, insulin resistance, waist circumference, and hypertriglyceridemia seem to be independent risk factors for NAFLD in hypothyroid or euthyroid patients [48]. Subclinical and/or overt hypothyroidism were associated with an increased risk of liver fibrosis [49,55] and low fT3 values were associated with increased liver stiffness and increased fibrosis score [57]. For the diagnosis of NAFLD, numerous studies have performed ultrasonography and thyroid dysfunction definition studies have used laboratory parameters, or/and history/self-reported of hypothyroidism on hormone replacement therapy. The majority of these studies were performed on the Asian continent. These results occurred after adjustments such as age, sex, smoking, body mass index, dyslipidemia, diabetes, hypertension, or for other cardio-metabolic risk factors were made. In the meta-analyses aforementioned, the authors adjusted multiple clinical and demographics variables. Guo et al. [60] documented that NAFLD individuals had higher TSH levels, while in Mantovani et al. [61], patients with primary hypothyroidism had a 2.7-fold higher risk of developing NAFLD/NASH. Rather, caution when reading the results of such studies, assumptions and hasty conclusions should not be made. However, the amount of evidence suggests that there is a significant association between THs derangement and NAFLD development, and strong consideration for such a possible disease connection should be made, especially if the patients suffer from obesity [63].

There are a few studies that did not discover any significant connection between hypothyroidism and liver steatosis [62,64,65,66,67,68,69,70,71]. Nonetheless, they pointed out that in women with NAFLD plasma, TSH concentrations were significantly higher than the controls [62], low fT4 levels were associated with liver steatosis [64], THs abnormalities could be due to sick euthyroid syndrome [65], hypothyroidism was associated with higher TG levels and a greater prevalence of obesity, and higher alteration of AST correlated with elevated TSH levels [71].

These inconclusive results emphasize the differences and limitations of studies represented by the study design, size, population demographics, environmental factors, the applied adjustments, definition, and diagnostic methods for NAFLD and hypothyroidism. Polymorphic variants [72], gender [17,73], and heterogeneity [74] are also variable factors that participate in HIN development. 

As seen, there is no clear evidence that may portray a cause–effect relationship between hypothyroidism and NAFLD, and the conflicting reports noted in the literature raises questions regarding this hypothesis. Further larger trials are needed to explore HIN, the molecular mechanism behind it, and maybe pave new roads regarding the use of thyromimetics in NAFLD.

## 2. Immunopathogenesis

Even if quondam investigations have thoroughly tried to describe the pathophysiology of hypothyroidism-induced NAFLD, the fundamental matrix behind this distinct entity is far from entirely known [24]. We know that low thyroid levels are commonly associated with hypometabolism. This state is defined by weight gain, reduced resting energy expenditure, reduced gluconeogenesis, and reduced lipolysis. THs dysfunction can instigate obesity, deficient lipid metabolism, and IR, which are elements of the metabolic syndrome also encountered in NAFLD [75,76,77]. Perhaps via the three-known mechanisms of NAFLD (hepatic lipid accumulation, inflammatory status accompanied by oxidative stress, and subsequently defective liver repair and regenerative response), hypothyroidism may directly or indirectly contribute to NAFLD. Thus, researchers have studied the profound interaction and signaling between THs, thyroid hormone receptors (TRs), and liver function [78,79]. 

### 2.1. Thyroid Hormone–Liver Axis

The thyroid gland is a pivotal endocrine organ responsible for thermogenesis, adipogenesis, fat distribution, energy, lipid, protein, carbohydrate, and cell metabolism [80]. The THs 3,5,3′-triiodothyronine (T3) and 3,5,3′,5′-tetraiodothyronine/thyroxine (T4) are important keys for tissue repair as they mediate cellular differentiation and interfere with cell-signaling mechanisms via protein–protein synergy through collaboration with nuclear receptors or binding to other proteins [81,82]. Production of THs from the thyroid gland is modulated centrally by thyrotrophs of the anterior pituitary that produce TSH, which in turn is regulated by the hypothalamus. The hypothalamus secretes the thyrotropin-releasing hormone (TRH). The secreted THs are then transported on cell membranes and their peripheral signaling is expressed in the adipose and hepatic tissue [83,84]. The TRs combine with another nuclear receptor and form the retinoid X receptor (RXR), which binds to thyroid hormone response elements (TREs) in regulatory portions of the target genes. The cytoplasmatic TRs regulate the noncanonical THs signaling, while the canonical signaling is considered the main mediator that controls, via TRβ, the hypothalamic–pituitary–thyroid axis. The TRs also control the actions of T3 [85,86,87]. There are two key thyroid receptor isoforms represented by the thyroid hormone receptor α1 and β1 (THR-α and THR-β). THR-α isoforms are ubiquitously expressed, but mainly found in the cardiovascular system, white adipose tissue (WAT), and in bone structures, whereas THR-β is mostly expressed in the hepatic organ and cardiac ventricle [85,88]. In NASH patients who underwent bariatric surgery, researchers found that THR-β messenger RNA (mRNA) negatively correlated with steatosis activity, implying that during disease development, there is a progressive resistance to THs. Resistance to TH is defined as a syndrome in which tissues have diminished sensitivity to TH [88]. THs exert different central and peripheral functions [89] and are influential modulators in liver regeneration as they can crosstalk with growth factors and integrins to regulate physio-pathological responses. THs mediate the number and function of different peroxisomal enzymes, and can regulate mitochondrial biogenesis and function in the hepatic cells (HCs) [90,91]. These particular properties of THs in the liver may explain the involvement of THs in liver cancer onset [92]. 

Tissue THs activity is controlled by deiodinases (D1, D2, and D3). Deiodinases are peroxidase seleno-dependent enzymes that are capable of activation or deactivation of THs in the peripheral tissues. Intrahepatic TH activity and homeostasis is regulated by serum TH levels and also by liver deiodinases [93]. While substrate T4 is converted into active hormone T3 by D1, D3 transforms T4 and T3 into inactive products, rT3, and diiodothyronine (T2), respectively. More than that, via D1, rT3 is catalyzed to T2 [94]. In healthy liver, hepatocytes express high levels of D1 while stromal cells express low D3 levels. Midst injury, a loss of D1 in hepatocytes is identified, while D3 expression in stromal cells increases, especially in fibrogenic myofibroblast. Intrahepatic hypothyroidism limits exposure to T3, which has an important role in cellular differentiation, is a greater regulator of metabolic systems than T4, and has oncosuppressor properties [95]. T3 expression is restrained during early phases of embryogenesis through D3 raised levels. After birth, D1 and D2 levels increase while D3 levels decrease, therefore T3 is granted the possibility to incite cellular differentiation and tissue maturation [96,97]. Evidence shows that after partial hepatectomy, myocardial infarction or in injured skeletal muscle, the expression of D3 is upregulated [98]. Additionally, in many chronic diseases, the hepatic D1 activity is reduced, whereas D2 along with TRs, uncoupling proteins, and beta-adrenergic receptors contribute to the development of obesity [99]. As a secondary mechanism for fat deposition, there is a raised conversion of T4 to T3 via enhanced deiodinase activity [100].

In liver injury, the expression of the nuclear TRs in HSCs is repressed, hence the main hormone receptor becomes THR-α, which starts an elaborate fibrogenic response and higher contractility. Detection in vitro of the HSC activation revealed that T3 may assist in the resolution of liver fibrosis in rats by mediating transforming growth factor-beta (TGF-β)-induced collagen I gene expression [101]. Injury-activated HSCs are crucial components for liver regeneration as they are major sources of myofibroblasts. These stromal cells promote mediators of angiogenesis, chemokines, and diverse growth factors [102]. As they activate, HSCs induce mutual changes in D1 and D3 hepatic expression, especially in D3, which diminishes the active-THs accumulation. These mechanisms can cross-talk in their way to affect TH expression [103]. Putative evidence is needed to elucidate the exact mechanism behind D1–D3 hepatic repair and the control behind TH homeostasis. The authors believe that D3 could represent a new therapeutic target in hepatic injury and fibrosis, and that rt3 could also be a novel biomarker [104]. This shows how intricate the relationship between the liver, THs, and TRs is and how different their influence is on each other, depending on the primary disorder.

### 2.2. Lipid and Cholesterol Metabolism 

Normally, there is an equilibrium between hepatic lipid synthesis and catabolism; however, an abundance of dietary intake leads to excess of plasma glucose, which enhances glycolysis and lipogenesis, with a subsequent rise in intracellular free fatty acids (FFAs) that can debilitate TR activity [105]. When energy intake happens, ATP is formed from carbohydrate oxidation and glycogen stocks are restored in the skeletal muscle and liver. Glycogen could account for the major deposit source of hepatic energy. Even if the link between liver glycogen and hypothyroidism is still unclear [106], the data show that low TH plasma concentrations may alter glycogen accumulation by decreasing the fatty acid synthase (FAS) and acetyl-CoA carboxylase (ACC) activity [107]. 

In excess conditions, the carbohydrate is transformed into fatty acids (FAs) via “de novo lipogenesis” (DNL). Most of the hepatic lipid accumulation is represented by re-esterification of circulating FFAs, followed by DNL, and finally, by dietary fatty acids. FAs go through one of the three metabolic transformations: β-oxidation (hepatic desaturation and elongation), esterification, and accumulation as TG or transformation and secretion as lipoproteins. NAFLD subjects have high levels of diacylglycerols, triacylglycerols, and elevated saturated FAs composition. Triacylglycerol can accumulate as fat droplets, or can be added to very-low-density lipoprotein (VLDL) or help with cellular reparation [108,109]. It seems that the derivative 3,5-diiodothyronine can regulate the activity of hepatic lipases to raise lipid mobilization from fat droplets [110]. DNL, along with triacylglycerol synthesis, are steadily coordinated by hormones and various crucial enzymatic processes [111]. THs regulate lipogenesis by binding to the specific genes of TRs, which can then mediate the transport of FFAs into the liver cells with the help of protein transports like liver fatty acid binding proteins (L-FABPs), fatty acid transporter proteins (FATPs), and fatty acid translocase (FAT). In the hepatocytes via THR-β, THs can increase the mitochondrial oxidation of FAs, promote the intrahepatic lipolysis through lipophagy with subsequently decreased TG clearance and increased TG hepatic up-take [112]. Studies show that TH enhances the activity and recruitment of Zinc-α2-Glycoprotein in hepatic cells that help facilitate lipolysis [113]. Nonetheless, this lipidic surplus can lead to cellular stress, apoptosis, and consequent liver impairment. Hence, the biochemical panel of NAFLD/NASH patients often register elevated liver transaminase, low-density lipoproteins (LDL), cholesterol, and TG levels [114]. 

Studies noted that NAFLD subjects have a lower concentration of THs [92], that small elevated TSH levels are linked to higher risk and prevalence of metabolic syndrome [115], and that TSH can hasten the onset and progression of NAFLD/NASH [116]. A sufficient reduction of TH levels is enough to lower the response of adipose tissue to adrenergic signaling. In these conditions, insufficient adrenergic stimulation of lipolysis in adipose tissue contributes by reducing the FA transport to the liver [117]. Additionally, THs stimulate the responses of catecholamine through enhancement of the expression of the uncoupling proteins in the mitochondria of skeletal muscle and obese cells by regulating the number of adrenergic receptors [16]. While TSH directly affects the hepatocyte cell membranes, it can promote hepatic lipogenesis, gluconeogenesis, and diminish hepatic bile acid synthesis [12,118]. Raised TSH levels were correlated with higher hepatic lipoprotein lipase activity [119,120]. These processes are mediated by THs through genomic and non-genomic mechanisms [121]. The first mechanism occurs with the help of the classical nuclear receptors THR-α and THR-β [120]. TSH promotes the TG hepatocyte up-take and endorse liver steatosis through binding to TSH receptor, which sets the hepatic sterol regulatory element-binding transcription factor 1 SREBP-1c activity via the cyclic AMP (cAMP)/protein kinase A(PKA)/peroxisome proliferator-activated receptor-α (PPARα) (cAMP/PKA/PPARα) pathway, paralleling with decreased AMPK function and subsequently raised lipogenesis. TSH can restrain cholesterol biosynthesis by stimulating AMPK-mediated phosphorylation of 3-hydroxy-3-methyl-glutaryl coenzyme A reductase (HMGCR) [122,123].

In addition, there is proof that highlights the indirect regulation of the liver lipid metabolism by the thyroid gland through the central nervous system [124]. TH indirectly drives the transcriptional regulation of liver lipogenesis as a result of its effect on carbohydrate-responsive element-binding protein (ChREBP) and liver X receptors. They can lower the VLDL, LDL, high-density lipoproteins (HDL), and apolipoprotein B100 levels [109]. TH can lower LDL by reducing the proprotein convertase subtilisin/kexin type 9 (PCSK9) levels. PCSK9 is controlled by SREBP-2, and data show that hypothroidism is associated with increased levels of PCSK9 [125]. Among other functions, TH instigates the expression of hepatic β-hydroxy β-methylglutaryl-CoA (HMG-CoA) reductase, which through different pathways, maintain constant sterol levels, and of cholesterol 7 alpha-hydroxylase (Cyp7A1) [62,118]. Lower CYP7A1 activity is correlated with higher LDL levels [72]. Experimental studies showed that a THR-β and T3 agonist can reduce Cyp7A1 levels and that Cyp7A1-TRs may represent a new therapeutic target [126]. Interestingly, the administration of synthetic THR-ß agonist (KB2115) reduced LDL serum levels [127]. Furthermore, the THR-β agonist (e.g., eprotirome, assobetirome) can lower circulating LDL-cholesterol levels without cardiac secondary effects [128]. A study on rodents showed that mice with a THR-α (ThrαPV/PV) mutation had lower hepatic lipid accumulation and decreased lipogenesis, while a negative mutation in THR-β (ThrβPV/PV) led to decreased fatty acid b-oxidation and raised PPARδ signaling, with subsequently increased lipid up-take and liver steatosis [129]. Additionally, THR-α knockout mice seem to be sheltered against diet-induced hepatic steatosis and peripheral IR [130]. 

THs perturb the regulation of HDL via enhancement of plasma protein factors like cholesteryl ester transfer protein (CETP), lecithin-cholesterol acyltransferase (LCAT), and hepatic lipase and decrease the synthesis of liver phospholipid (phosphatidylserine and cardiolipin) and sphingolipid species [131]. Dysregulations in this protein metabolism act in synergy with the severity of hypothyroidism [132]. THs increase cholesterol efflux from peripheral tissues through the enhancement of genes such as Apolipoprotein A1 (Apo A1) and scavenger receptor class B member 1 (SRB1) [133]. Another complex mechanism by which THs can modulate lipid metabolism are represented by transcriptional, non-transcriptional, and microRNA mechanisms [134]. For example, via miR181d, THs decrease and import a transcription factor that can modulate the activity of sterol O-acyltransferase 2 (SOAT2), which has a crucial role in the conversion of cholesterol to specific esters [135]. T3 may have protective effects against lipotoxic derivatives and its action is regulated via mRna and the activity of liver lipases. Additionally, T3 via hepatic lipophagy transports lipids to lysosomes and enhances FA oxidation, which promotes long term hepatic cell autophagy and damage [136,137]. Previous treatment with a high dose of the T3 hormone led to a decrease in cholesterol levels and promoted weight loss, however, its use was limited because of its serious cardiac side effects [138].

These processes emphasize the hepatic lipidic balance and wealth that is induced and sustained by THs. It is clear that lipidomic signatures have a decisive part in NASH evolution/resolution [139,140], that lipid and cholesterol metabolism alteration is beyond complex, and represent the hallmarks of NAFLD; however, supplementary and sustainable evidence to describe these changes in hypothyroidism and how exactly they connect with TH levels are desired.

### 2.3. Glucose and Insulin Metabolism

Hyperinsulinemia as well as hypothyroidism were reported to contribute to NAFLD development independently of each other [140]. In reduced TH plasma levels, the β-pancreatic cell sensitivity for glucose is damaged, which leads to reduced insulin secretion. Hence, the lipolysis is impaired postprandial and the transport of FFAs from the adipose tissue to the liver and vice versa increases [141]. Hepatic lipid surplus enhances IR, which leads to the deficient suppression of postprandial glucose production. Furthermore, the esterification of FAs and uptake of TG in the liver are supplied by the hepatic IR [19,142]. Glucose homeostasis is impaired by reduced insulin secretion and TG accumulation, which alters the insulin-mediated clamp decreased endogenous glucose production. The raised plasma glucose controls the hepatic DNL [115]. Studies showed that smoking and IR may impact the actions of subclinical hypothyroidism on the lipid profile [143], and that defective insulin mediated lipolysis is suppressed in mild hypothyroidism, but not in severe hypothyroidism [115]. It was reported that insulin resistance is associated with hypothyroidism and that IR sensitivity improves in levothyroxine treated patients [144]. Accentuated hyperglycemia found in hypothyroidism could explain the connection to IR, diabetes onset, and NAFLD development [145].

### 2.4. Oxidative Stress and Inflammation

Oxidative stress and inflammation are one of the items that unlock and sustain many cardiovascular, endocrinological, and liver diseases, hence, early evaluation of markers that indicate their presence should be evaluated. The inflammatory pattern in NAFLD/NASH is usually represented by high TNF-α, interleukin-6 (IL6), and chemokine levels [146]. Along with an accumulation of hepatic FFAs and inflammation, malfunction of the mitochondria appears. The altered mitochondria, also called the “powerhouse” of the cell [147], leads to excessive oxidation and overflow of reactive oxygen species (ROS). These ROS products impair the activity of deiodinases [148], enhance lipid peroxidation, activation of the KCs, and of specific pro-inflammatory cytokines. Among them, the TNF-α and transforming growth factor β (TGF-β) activate hepatic satellite cells and endorse fibrosis [149,150]. THs modulate the hepatocyte mitochondrial function by the uncoupling of oxidative phosphorylation [151], and generate ROS through Ca^2+^–calcium/calmodulin-dependent protein kinase kinase 2 (CAMKK2)–5′-AMP-activated protein kinase (AMPK) circuit. The hepatic autophagic removal of mitochondria and their biogenesis are both modulated by THs. More than that, by activating Ca^2+^–AMPK and the cAMP–protein kinase A (PKA) pathways, THs can mediate lipid metabolism [152].

In hypothyroid rodents, researchers have found an inflammation and decreased level of ceramides, which account for decreased cell differentiation and apoptosis [153]. Hashimoto’s thyroiditis patients with hypothyroidism have increased serum markers of oxidative stress (e.g., malondialdehyde) [154]. Metabolic products of inadequate fatty acid oxidation set off and worsen liver inflammation, fibrosis, and necrosis [155,156]. Recently the 3,5-diiodo-L-thyronine (3,5-T2) molecule has attracted more attention because of its metabolic effects, which are similar to T3, but mainly because of its beneficial actions on mitochondrial function and oxidative stress, which suggest its future introduction as a potential clinical drug [157]. More data are needed regarding inflammation and oxidative stress in HIN, which may bring interesting possibilities of new therapeutic targets or biomarkers.

### 2.5. Adipokines and Hormones

One of the mechanisms possibly involved in the hypothyroidism-NAFLD pathogenesis could be represented by adipokine metabolism. Low circulating levels of thyroid hormones influence certain adipocytokines levels such as adiponectin or leptin. Adiponectin can enhance fatty acid oxidation and inhibit DNL through activation of AMP-activated protein kinase [158,159]. Hypothyroid subjects have altered adiponectin levels that can contribute to IR development. The modified adipocytokines registered have hepatotoxic properties, can promote oxygen radical release, and furnish liver inflammation and fibrosis [160,161]. Leptin is one hormone that is in charge of appetite modulation, can promote hepatic collagen synthesis hepatic IR, and is involved in hepatic fibrogenesis [162]. High levels of this hormone were found in patients with thyroid disfunction and seems to be correlated with TSH levels and with BMI [163]. Another adipokine studied in NAFLD and hypothyroidism is visfatin. In NAFLD, and especially in NASH subjects, low levels of visfatin have been described, however, an exact explanation behind this association is still unclear [164]. Recently, it was proposed that visfatin could be a promising serum biomarker for monitoring liver disease in younger obese population [165].

In addition to the hormonal branches that are explored in HIN pathogenesis, we mention fibroblast growth factor-21 (FGF-21). FGF-21 is a pleiotropic hormone, and specifically a member of the FGF endocrine subfamily, which expresses hormone-like activities and is considered a major regulator of energy homeostasis [166]. Previously, it was described that NAFLD patients have higher FGF-21 levels compared to heathy subjects and that this can fasten NAFLD progression [167]. More than that study shows that hypothyroid subjects have increased levels of FGF-21 independently of lipid profile [168]. Experimental studies have shown that administration of FGF-21 led to improved glucose clearance, insulin sensitivity, and scaled down TG levels [169]. Few studies investigated how hormonal changes influence HIN. Menopause can induce changes in hormonal status and lipid metabolism. Reduced estrogen production leads to increased plasma cholesterol levels, these changes being similar to those encountered in overt hypothyroidism. Recently, the SardiNIA study found that a postmenopausal status influences the relationship between lipid profile and TSH levels [170]. The particular role of adipokines and other hormones such as estrogens in HIN need further exploration.

There is a fine balance between certain key elements found in metabolic syndrome that are also common seen in NAFLD and/or hypothyroidism. This sophisticated relationship, which incorporates alteration in the lipid, TG, cholesterol metabolism, IR, inflammation and oxidative stress, and the dysfunction of the hepatic hormone, is by some degree shaped by THs (Figure 1). We hope that with a better understanding of HIN pathophysiology, advanced technology can help us find new therapeutic approaches. 

## 3. Hypothyroidism-Induced NAFLD Treatment

Considering the burden of NAFLD and that no current licensed therapy exists, THs and their metabolites, along with the THR-β agonist and other liver specific analogs have been tested in the last few years as a potential NAFLD therapy [7,59,60,171].

### 3.1. Thyroid Hormones

The synthetic form of the T4 hormone is levothyroxine sodium, which contains crystalline L-3,3′,5,5′-tetraiodothyronine sodium salt, having the exact chemical structure of that produced in the human body by the thyroid gland. T3 is the thyroid hormone that produces most of the physiological actions. It is derived mostly from T4 (approximately 80%) by deiodination in different tissues. THs bind to the DNA receptor proteins after they diffuse into the nucleus and activate gene transcription along with messenger RNA synthesis and production of cytoplasmic proteins. Levothyroxine is absorbed mostly in the gastrointestinal tract (in the jejunum and upper ileum). The level of absorption varies from 40% to 80%, is increased by fasting, and decreased in malabsorption syndromes. Circulating THs bind to plasmatic proteins like albumin (TBA), thyroxine-binding prealbumin (TBPA), and thyroxine-binding globulin (TBG), but are metabolically active only unbound. The main degradation site for the THs is the liver, but T4 deiodination to T3 and reverse T3 (rT3) also occurs in the kidneys and other tissues. Diiodothyronine is the result of T3 and rT3 deiodination. THs are mostly eliminated through urinary excretion and we have to be aware that it decreases with age [172]. Levothyroxine is the principal treatment for hypothyroidism. It is also associated with a decrease in serum lipids and body mass index (BMI) [77].

In a study published in the *International Journal of Endocrinology* in 2017, the effect of levothyroxine (LT_4_) treatment on NAFLD resolution was confirmed in patients with subclinical hypothyroidism (SCH). There were two main subgroups: 33 patients with significant SCH (TSH of at least 10 mIU/L) and 330 with mild SCH (TSH between 4.2 and 10 mIU/L). The second group was further divided into one group that received LT_4_ treatment (181 patients with mild SCH: mild SCH-LT_4_) and another one that did not (149 patients representing the control group: mild SCH-control group) [173,174]. All those with significant SCH received LT_4_ treatment (median LT_4_ dosage in this group was 75 µg per day). Those in the mild SCH-LT_4_ group received a median LT_4_ dosage of 50 µg per day [175]. After a 15-month follow-up, the prevalence of NAFLD was reduced in the significant SCH group from 48.5% to 24.2% (*p* = 0.041). There was a moderate reduction in the prevalence of NAFLD in mild SCH patients that received LT_4_ treatment, but it was not statistically significant (from 44.2% to 35.9% with a *p* value of 0.108). In the mild SCH-control group, there was also a reduction in the NAFLD prevalence from 39.6% to 34.9% (*p* = 0.402). These results show that LT_4_ supplementation can be effective in NAFLD reduction if it is used in patients with significant SCH. There was also a decrease in serum AST value of 5.61 IU/L (*p* < 0.001) in significant SCH patients after LT_4_ treatment, but not a statistically significant one in serum alanine aminotransferase (ALT). LT_4_ supplementation in the mild SCH group was associated with a reduction in both AST and ALT, with the last one being marginally significant (from 19.09 IU/L to 17.95 IU/L, *p* = 0.087). In the mild SCH-control group, serum ALT was stable, but there was a reduction of serum AST. There was a remarkably greater reduction of serum AST in mild SCH-LT_4_ patients compared to the mild SCH-control group (*p* = 0.046). Dyslipidemia is also considered to be a risk factor for NAFLD. A subgroup analysis was performed on mild SCH patients that also had dyslipidemia (207 out of 330). The results showed that those that underwent LT_4_ treatment had higher reductions in NAFLD prevalence and serum liver enzyme values compared to those in the control group. This confirms the strong connection between thyroid function and NAFLD [173,174].

In a 2018 phase IIb trial that took place in six hospitals in Singapore, a significant reduction of hepatic fat was confirmed by magnetic resonance spectroscopy after 16 weeks of low-dose levothyroxine administration. The study included 20 euthyroid patients who had already been diagnosed with type 2 diabetes and NAFLD [175].

T3 (3,5,3′-triiodo-L-thyronine) is known to regulate numerous physiological processes, with one of the most targeted organs being the liver [176]. Exogenous administration of T3 after a choline-methionine deficient (CMD) diet that induces hepatic triglycerides accumulation decreased liver steatosis [177]. T3 promoted fatty acid peroxisomal and mitochondrial β-oxidation. Furthermore, it decreased hepatic fatty acid-binding protein (L-FABP) expression [178]. In addition, lipid peroxidation and expression of cyclooxygenase-2 (COX-2) were reduced. There was a noted decrease in phosphor-stress-activated protein kinase/c-Jun NH2-terminal kinase (phosphor-SAPK/JNK) and phosphor-signal transducer and activator of transcription 3 (phosphor-STAT3) levels [179,180]. Cable and co-workers demonstrated on rats that T3 reduces the hepatic mRNA levels of apolipoprotein C3 (ApoC3) and sterol regulatory element binding protein-1c (SREBP-1c). Moreover, it increases peroxisome proliferator-activated receptor γ coactivator-1α (PGC-1α) and apolipoprotein A1 (ApoA1) levels [181]. According to a recent study conducted on high fat diet (HFD) rats, T3 increases carnitine palmitoyltransferase-1 (CPT-1) levels with a significant decrease in liver fat content [131]. Other studies have suggested that THs induce hepatic autophagy in order to provide the mitochondria with fatty acids [182]. In a 2014 study performed on hypothyroid mice, T3 restored the level of saturated fatty acid. In addition, the expression of acetyl-CoA carboxylase 1 and enzymes used in de novo lipogenesis was increased after T3 treatment. Stearoyl-CoA desaturase-1 activity was suppressed by TH administration, suggesting that they might decrease triglyceride accumulation in the liver [106]. Furthermore, there is growing interest in the possible T3 central effects on the liver. Alvarez-Crespo observed in a 2016 study how T3 chronic central infusion influenced adipose tissue activity and thermogenesis [183]. Unfortunately, increased T3 levels are known to have cardiac, muscle, and bone harmful effects [184].

### 3.2. Thyroid Hormone Metabolites

In the last few decades, T2 (3,5-diiodo-L-thyronine) has emerged as a bioactive TH-related compound. Most in vitro experiments suggest that T2 results after T3 deiodination, having a significantly lower affinity for THRs compared to T3. There are three main selenoenzymes that trigger deiodination, out of which type 2 deiodinase (D2) seems to be the one involved in T2 formation [185,186]. Both in vitro and in vivo models observed that T2 has significant effects on the reduction of hepatic lipid accumulation resulted after exposure to HFD. It also had an impact on the reduction of acyl-CoA oxidase (AOX) activity and peroxisomal β-oxidation [187]. Further studies suggested that T2 acts directly by activating hepatic nuclear sirtuin 1 (STIR1) [188]. Recently, in a 2017 study performed on HFD fed rats, the effects of T2 on hepatic metabolism were observed in parallel to those of T3. They both decreased hepatic triglyceride levels and induced liver autophagy along with intra-hepatic acylcarnitine accumulation. In addition, they prevented sphingolipid-ceramide generation, but only T2 rescued the impairment caused by HFD in protein kinase B (AKT) and mitogen-activated protein kinase/extracellular signal-regulated kinase (MAPK/ERK) [131]. T2 achieves the inhibition of liver fat accumulation by decreasing lipogenesis and increasing fatty acid oxidation [188]. Used at a higher dose, T2 had potential cardiac side effects [178]. A functional analog of T2 (TRC150094, TRC) also had an impact in reducing hepatic triglyceride content in HFD rats by increasing oxidation and mitochondrial fatty acid import, without any undesirable effects [189]. Type 3 deiodinase (D3) generates rT3 from T4 and 3,3′-diiodo-L-thyronine (3,3′-T2), another form of T2, from T3. A higher affinity for THRs was observed for 3,3′-T2 in HFD-fed mice, but had a significant negative impact on metabolic parameters [186,190].

Another TH metabolite is 3-iodothyronamine (T1AM), which does not bind to THR, but mediates its effects through the trace amine-associated receptor (TAAR1) [191]. Recent studies demonstrated that T1AM influences body weight and different metabolic pathways that are usually associated with NAFLD [192]. In addition, T1AM reaches higher concentrations at the hepatic level, which is why it is considered that its metabolic effects are partly mediated by hepatocytes [193]. HepG2 cells had a significant T1AM uptake, with an increase in glucose production. There was also an increase in the production of ketone bodies at the hepatic level, which might be correlated with pyruvate and amino acid catabolism [194]. Furthermore, in normal hepatocytes, a direct inducing fatty acid catabolism was observed. Chronic T1AM administration was correlated with a lipolytic pattern, confirmed by in vivo studies that associated significant weight loss [195]. Assadi-Porter and colleagues demonstrated in a 2018 study that the increased fat oxidation observed at a hepatic level comes not only from elevated ketone bodies (3-hydroxybutyrate and acetone), but also from carnitine and succinate. Furthermore, they determined that T1AM effects on lipid hepatic metabolism were dose-dependent [196].

### 3.3. Alternative Treatment

In a recent study published in 2019, thymoquinone (TQ) is used for restoring the liver structure to its initial state after induced hypothyroid NAFLD. TQ is the main constituent of *Nigella salvia* (NG), a medicinal plant representing the *Ranunculaceae* family [197]. NG is mostly used for its antioxidant and anti-inflammatory properties, but it also has an immunomodulatory effect [198]. Experimental hypothyroidism was confirmed by the reduction of T3 and T4, associated with a significantly increased TSH in the serum of rats that were given 6-propyl-2-thiouracil (PTU) for two weeks. After that, they were divided into a group that continued PTU administration and another one that had concomitant administration of PTU and TQ for the next four weeks. There was also a control group and one that received TQ for six weeks. The livers of the rats were processed for obtaining paraffin blocks and Hematoxylin and Eosin and Streptavidin-Biotin-Peroxidase staining were used. The structure and general architecture of the livers were slightly affected and those of the group treated with PTU showed fatty degeneration (steatosis) in the form of microvesicles and macrovesicles within the hepatocytes. Additionally, alongside classical fatty lesions, intralobular inflammatory reaction was observed. In the group treated with PTU + TQ, resolution of the steatotic lesions and that of steatohepatitis was observed. There was also a decrease in NAFLD activity scores and lobular inflammation. Advanced fibrosis was detected through alpha-smooth muscle actin (α-SMA) staining [199]. A significant increase in the α-SMA reaction in the livers of the PTU treated group was observed mainly in the portal region associated with focal inflammatory response areas. The α-SMA index was restored to that of the control group after TQ administration. α-SMA staining was also used in the detection of activated hepatic stellate cells (HSCs) [200], which is a key element in hypothyroidism induced NAFLD. Anti-CD68 antibodies were used to evaluate the number of hepatic macrophages (Kupffer cells). It is known that Kupffer cells are activated by liver injury, which leads to HSC activation and fibrogenesis [201]. There was an important increase in the portal and intralobular CD68^+^ cells in the PTU treated group compared to the PTU + TQ and control groups. The main defense mechanism of the liver against the harmful effect of reactive oxygen species (ROS) is exerted by the catalase (CAT) enzyme. It was demonstrated by Subudhi and Chainy that CAT regulation is one of the functions of thyroid hormones [202]. Upregulation of CAT gene expression at a hepatic level in PTU rats that were treated with TQ demonstrated the antioxidant effect of TQ that can annihilate the oxidative stress effect of hypothyroidism on the liver [203].

## 4. THR-β Selective Thyromimetics and NAFLD

THs have affinities for both THR-α and THR-β, this is why their use is limited due to potential adverse reactions that result mostly from THR-α activation [204]. Molecules targeting specific thyroid hormone receptors have been tested. The main thyroid hormone receptor expressed in hepatocytes that regulate metabolic pathways involved in the pathogenesis of NAFLD is THR-β [182]. Triiodothyronine and thyroxine have beneficial metabolic effects on cholesterol and triglycerides by targeting the hepatic THR-β. The first molecules of THR-β-selective agonists were sobetirome (GC-1) and eprotirome (KB-2115) [103]. In a 2008 study performed on rats, the effects of T3 and GC-1 (a synthetic TH analog that binds to THR-β1 with the same affinity as T3) on CMD diet-induced NAFLD/NASH were analyzed. Both T3 and GC-1 prevented hepatic fat accumulation by increasing fatty acid β-oxidation. In addition, there was hepatosteatosis regression due to a decrease in lipid peroxidation and expression of COX-2, associated with phospho-STAT3 and phospho-SAPK/JNK activation [179]. Further studies showed that GC-1 did not have the requisite safety profile causing insulin resistance and hyperglycemia, so it has not been used as a treatment in NAFLD [103,204]. KB-2115 was tested in a 2013 study on fat-fed rats and had statistically significant results in reducing hepatic steatosis, but also decreased GLUT4 skeletal muscle content [204]. In a 2015 parallel study for sobetirome and eprotirome, they both had a significant effect in lowering cholesterol serum levels, but eprotirome long-term use resulted in cartilage disruption in dogs [205]. Another orally active THR-β agonist used to reduce hepatic steatosis in a 2009 study on rats and mice is MB07811. It was demonstrated that it increases β-oxidation and mitochondrial respiration rates, reducing hepatic triglycerides [156]. Most THR-β-selective agonists had important adverse effects and the studies did not go any further, even though they had a major role in reducing the circulating lipids [103]. 

The latest selective thyroid hormone β-receptor agonist is resmetirom (MGL-3196). It is orally active and liver-directed with a higher selection rate than triiodothyronine for THR-β compared to thyroid receptor α (THR-α) [206]. MGL-3196 also has a very high protein bound (above 99%) with very good hepatic tissue penetration, showing specificity for the liver. Testing of resmetirom on animal models showed that it has an important role in reducing hepatic triglycerides, lipid peroxidation, ALT, steatosis, inflammation, and fibrosis [157,205]. In a 36-week randomized study that was double-blinded and placebo-controlled and took place in 25 USA centers, only adults with biopsy confirmed NASH (stages 1–3 of fibrosis) were enrolled and the eligible ones were those who had a hepatic fat fraction above 10% at MRI-proton density fat fraction (MRI-PDFF). Patients that had associated hypothyroidism with thyroxine treatment doses >75 μg daily were not included in the study. A history of alcohol consumption, drug induced NAFLD, uncontrolled type 2 diabetes (the value of glycated hemoglobin ≥ 9.5%), or use of glucagon-like peptide analog (unless they were using it for more than six months before screening on a stable dose) were also exclusion criteria. In addition, patients with severe hepatic dysfunction (hepatic decompensation, cirrhosis, or other chronic liver diseases) and with elevated (five times more the upper limit of normal range) serum ALT or aspartate AST were not included. This phase 2 study used an atypical dosing regimen by administrating 80 mg per day with a possible dose adjustment (± 20 mg/day) after four weeks. The study concluded that resmetirom (administrated daily in oral doses) had a significant impact on the reduction of the hepatic fat fraction (measured by MRI-PDFF). The relative liver fat reduction after 12 weeks of treatment was 26.7% compared to the baseline (placebo adjusted), associated with a relative reduction in ALT of 5.8% from the baseline (placebo adjusted). There was a sustained reduction in hepatic fat after 36 weeks of treatment (29% relative reduction from baseline, placebo adjusted) [207]. In addition, resmetirom therapy had a notable impact on the reduction of atherogenic lipids and lipoproteins such as triglycerides, LDL cholesterol, apolipoproteins B and CIII, and lipoprotein(a) [207]. There were improvements in the levels of liver injury markers and fibrosis, with reduced features of NASH on liver biopsy in 27% of those who underwent resmetirom therapy compared with 6% of patients that received placebo [208,209]. One of the mechanisms through which resmetirom reduces liver fat is thought to be the increase of mitochondrial β oxidation [157]. It is also assumed that resmetirom reduces VLDL production and secretion, lowering LDL cholesterol, triglycerides, and apolipoprotein B plasma concentrations. The mechanism through which MGL-3196 reduces serum concentration of lipoprotein(a) is still unknown [210]. Another THR-β selective agonist, VK2809 (MB 07811), was used in a 12-week phase 2 study as a treatment for patients with NAFLD. The subjects received 10 mg of VK2809 per day and at the end of the study, there was a 50.8% liver fat reduction from the baseline (placebo adjusted) [211]. In a 2016 study, a hybrid molecule composed of glucagon and T3 was tested, targeting synchronized signaling in order to minimize the adverse effects of each hormone. Glucagon/T3 conjugates had a beneficial impact in reducing hepatic fat in metabolically compromised mice, without cardiovascular or bone toxicity. These findings open a new era for chronic treatment in the vast number of pathologies that constitute what we call metabolic syndrome including NAFLD [212]. Based on the available evidence (Table 1), we conclude that further research should be focused on this issue, especially given the prevalence of NAFLD associated with hypothyroidism.

## 5. Conclusions

Robust evidence is needed to delineate the nature behind NAFLD-hypothyroidism pathogenesis, how they influence each other, and if, in the near future, a program of detecting the thyroid function in NAFLD patients would be beneficial. As NAFLD specific treatment is still unavailable and its management relies most on lifestyle changes, we may not be as reluctant to test thyroid parameters in these patients. Of interest is thyroid replacement therapy that could improve disease progression and lead to better outcomes. New larger clinical trials are needed to assess sufficient evidence that support and complement the existing hypothesis, and perhaps find the best therapy options for these two combined diseases. 

## Figures and Tables

**Figure 1 ijms-21-05927-f001:**
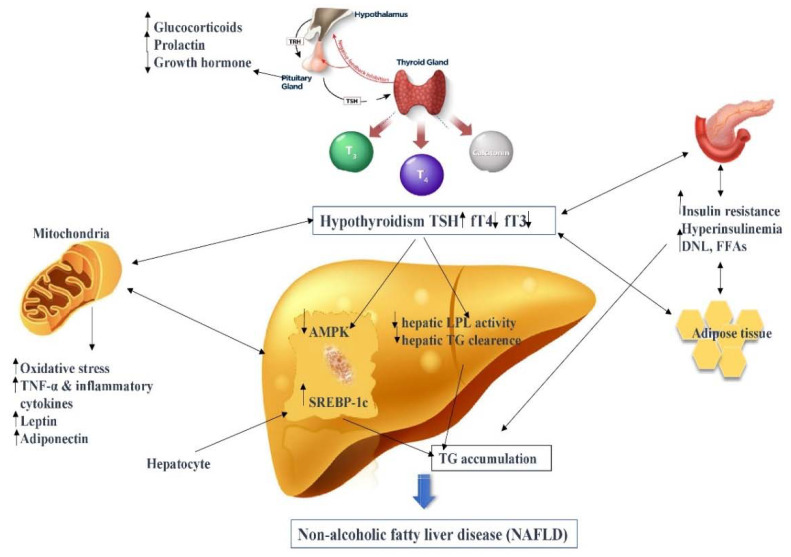
The complex relationship of hypothyroidism-induced NAFLD (HIN). Thyroid stimulating hormone (TSH); free thyroxine (fT4); free triiodothyronine (fT3); de novo lipogenesis (DNL); free fatty acids (FFAs); lipoprotein lipase (LPL); triglycerides (TG); AMP-activated protein kinase (AMPK); sterol regulatory element-binding protein (SREBP-1c); tumor necrosis factor alpha (TNF-α).

**Table 1 ijms-21-05927-t001:** The impact of TH therapy on liver function in NAFLD. Nonalcoholic fatty liver disease (NAFLD); 6-propyl-2-thiouracil (PTU); nonalcoholic steatohepatitis (NASH); thyroid hormone receptor β (THR- β); low-density lipoproteins (LDL); choline-methionine deficient (CMD); cyclooxygenase-2 (COX-2); phosphor-signal transducer and activator of transcription 3 (phospho-STAT3); phosphor-stress-activated protein kinase/c-Jun NH2-terminal kinase (phospho-SAPK/JNK); high fat diet (HFD); protein kinase B (AKT); mitogen-activated protein kinase/extracellular signal-regulated kinase (MAPK/ERK); carnitine palmitoyltransferase-1 (CPT-1); messenger RNA (mRNA); apolipoprotein C3 (ApoC3); apolipoprotein A1 (ApoA1); sterol regulatory element-binding transcription factor 1 (SREBP-1c); peroxisome proliferator-activated receptor γ coactivator-1α (PGC-1α); liver fatty acid binding protein (L-FABP); levothyroxine (LT_4_); subclinical hypothyroidism (SCH).

	Country	Year	Main Findings	NAFLD Impact	First Author (Reference)
**Thyroid hormones**					
Levothyroxine (T4)	China	2017	LT4 replacement therapy had beneficial impact in patients with significant SCH or mild SCH with dyslipidemia	NAFLD reduction in patients with significant SCH	Liu, L. [174]
	Singapore	2018	low-dose LT4 therapy decreased intrahepatic lipid content in 20 euthyroid male patients diagnosed with type 2 diabetes	significant reduction of hepatic fat in a small group of patients	Bruinstroop, E. [175]
Triiodothyronine (T3)	Italy	2008	promoted fatty acid peroxisomal and mitochondrial β-oxidation decreased L-FABP expression reduced lipid peroxidation and COX-2 expression decreased phosphor-SAPK/JNK and phosphor-STAT3 levels	decreased CMD diet-induced hepatosteatosis in rats prevented hepatic fat accumulation by increasing fatty acid β-oxidation	Perra, A. [179]
	USA	2009	reduced the hepatic mRNA levels ApoC3 and SREBP-1c increased PGC-1α and ApoA1 levels	induced adipocyte lipolysis reduced ability to decrease hepatic steatosis	Cable, E.E. [181]
	China	2014	restored the level of saturated fatty acid increased expression of acetyl-CoA carboxylase 1 and other enzymes used in de novo lipogenesis	stearoyl-CoA desaturase-1 activity was suppressed by TH administration, suggesting that they might decrease triglyceride accumulation in the liver	Yao, X. [106]
	Spain	2016	chronic central infusion influences adipose tissue activity and thermogenesis	-	Alvarez-Crespo, M. [183]
	Italy	2017	increased CPT-1 levels	significant decrease in liver fat content in HFD-fed rats	Iannucci, L.F. [131]
**Thyroid hormones metabolites**					
T2 (3,5-diiodo-L-thyronine)	Italy	2017	reduction of acyl-CoA oxidase activity and peroxisomal β-oxidation	reduction of hepatic lipid accumulation resulted after exposure to HFD	Grasselli, E. [187]
	Italy	2017	prevented sphingolipid-ceramides generation rescued the impairment caused by HFD in AKT and MAPK/ERK	decreased hepatic triglyceride levels and induced liver autophagy along with intra-hepatic acylcarnitine accumulation in HFD fed rats	Iannucci, L.F. [131]
	Italy	2017	decreased lipogenesis increased fatty acid oxidation	inhibition of liver fat accumulation	Senese, R. [188]
T1AM (3-iodothyronamine)	Italy	2014	T1AM reached higher concentrations at hepatic level increased the production of ketone bodies at hepatic level	-	Ghelardoni, S. [193]
	USA	2018	increased fat oxidation observed at a hepatic level comes also from carnitine and succinate	dose-dependent effects on lipid hepatic metabolism	Assadi-Porter, F.M. [196]
**THR-β Selective Thyromimetics**					
Sobetirome (GC-1)	Italy	2008	decreased lipid peroxidation and expression of COX-2, associated with phospho-STAT3 and phospho-SAPK/JNK activation	decreased CMD diet-induced hepatosteatosis in rats prevented hepatic fat accumulation by increasing fatty acid β-oxidation	Perra, A. [179]
Eprotirome (KB-2115)	USA	2013	decreased GLUT4 skeletal muscle content	significant reduction of hepatic steatosis in fat-fed rats	Vatner, D.F. [204]
	USA	2015	long term use resulted in cartilage disruption in dogs significant effect in lowering cholesterol serum levels	-	Lammel Lindemann, J. [205]
MB07811	USA	2009	increased β-oxidation and mitochondrial respiration rates, reducing hepatic triglycerides	reduction of hepatic steatosis in rats and mice	Cable, E.E. [181]
Resmetirom (MGL-3196)	USA	2019	reduced atherogenic lipids and lipoproteins, such as triglycerides, LDL cholesterol, apolipoproteins B and CIII, and lipoprotein(a)	significant impact on the reduction of the hepatic fat fraction	Harrison, S.A. [207]
VK2809 (MB 07811)	USA	2018	further development as a low-dose THR-β selective thyromimetic for NASH patients	50.8% liver fat reduction from baseline in NAFLD patients	Loomba, R. [211]
**Glucagon/T3 Conjugate**					
	Germany	2016	no cardiovascular or bone toxicity	beneficial impact in reducing hepatic fat in metabolically compromised mice	Finan, B. [212]
**Thymoquinone**					
	Saudi Arabia	2019	antioxidant, anti-inflammatory properties and immunomodulatory effect	restored the liver structure to its initial state after PTU-induced hypothyroid NAFLD	Ayuob, N.N. [197]

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
