# Peer review of "Hypothyroidism-Induced Nonalcoholic Fatty Liver Disease (HIN): Mechanisms and Emerging Therapeutic Options"

_ijms, 2020, doi:10.3390/ijms21165927_

Round 1

Reviewer 1 Report

General Comment

This is an intriguing model of liver disease which recognizes a definite and potentially curable cause. However, the present study faithfully mirrors rationale and structure of similar reviews, that have been cited. Among the limitations: English must carefully be edited by a native speaker.

Specific comment

Throughout the manuscript rephrase non-alcoholic--> nonalcoholic

Lines 49-51 Genetics must be mentiones

Line 52 Again, hereditary conditions as well as autoimmune, drug-induced liver injury must be mentioned.

54 - "Rapid evolution" not invariably. This statement should be softened.

63 - "Debilitated" --> below range

64 - Linked to, atrial fibrillation

66 - continues

73 - please, note that the first proposal to identify HIN as a distinct disease entity is NOT ref 24 but ref. 78

76-77 - Reword as follows: "growing epidemiological evidence associating these two diseases"

78 - Newly present (?)

79-81 Thi statement should be referenced (e.g. Lonardo, IJMS)

94-95 sentence unclear

96-98 Ref ?

99 - "many side effects" I would soften this statement

109 show

182 commonly

251 use and distribution--> synthesis and catabolism ?

372 snf 388 hypothyroid

380 enthuzistimatic --> ?

384 possibly

422 - proper --> licensed

631 gastro--> hepato

631 colossal --> increased

632 sentence unreferenced, please either delete or soften

Reviewer 2 Report

This review by Tanase et al is focused on the association between hypothyroidism and NAFLD.

The aim is interesting but the article is too long and I suggest to short. In addition references are too much for a non-systematic review, most of which are repetitive.

1) Line 64: atrial fibrillation?

2) Chapter 1.1 I suggest to change the title as follow: definitions and epidemiological discoveries.

3) A recent studies found that menopause modulates the association between thyroid hormone and lipids. Authors should give a short comment.

4) Revision of English language
